# Involvement of the AMPK Pathways in Muscle Development Disparities across Genders in Muscovy Ducks

**DOI:** 10.3390/ijms251810132

**Published:** 2024-09-21

**Authors:** Wanxin Zhao, Yulin He, Ziyuan Du, Xuanci Yu, Juan Chen, Ang Li, Caiyun Huang

**Affiliations:** College of Animal Science, Fujian Agriculture and Forestry University, Fuzhou 350002, China; 15509296551@163.com (W.Z.); heyulin0813@163.com (Y.H.); 13266460005@163.com (Z.D.); 15611021795@163.com (X.Y.); chenjuan78413@126.com (J.C.); liang@sina.com (A.L.)

**Keywords:** muscle, growth traits, hormone, AMPK

## Abstract

The differences in muscle development potential between male and female ducks lead to variations in body weight, significantly affecting the growth of the Muscovy duck meat industry. The aim of this study is to explore the regulatory mechanisms for the muscle development differences between genders. Muscovy ducks of both sexes were selected for measurements of body weight, growth traits, hormone levels, and muscle gene expression. The results show that male ducks compared to females had greater weight and growth traits (*p* < 0.05). Compared to male ducks, the level of serum testosterone in female ducks was decreased, and the estradiol levels were increased (*p* < 0.05). The RNA-seq analysis identified 102 upregulated and 49 downregulated differentially expressed genes. KEGG analysis revealed that among the top 10 differentially enriched pathways, the AMPK signaling pathway is closely related to muscle growth and development. Additionally, the mRNA and protein levels of CD36, CPT1A, LPL, and SREBP1 were increased and the P-AMPK protein level decreased in the female ducks compared to the male ducks (*p* < 0.05). In conclusion, muscle development potential difference between male and female ducks is regulated by sex hormones. This process is likely mediated through the activation of the AMPK pathway.

## 1. Introduction

China is the world’s leading region for duck meat production, accounting for 69% of the global share [1]. The Muscovy duck is one of China’s main meat duck breeds, known for its tolerance to rough feed, strong disease resistance, rapid growth, and high meat yield [2,3]. Notably, the weight difference between male and female Muscovy ducks at 10 weeks of age can be as much as 3 to 4 times. At 10 weeks, male ducks weigh about 3 to 5 kg, while female ducks weigh about 1.8 to 2.25 kg [4]. Female ducks grow more slowly and have lower lean meat rates and smaller body weights, which severely impacts the meat value of Muscovy ducks and the development of the industry.

As the main meat part, the difference in muscles between male and female Muscovy ducks is even more pronounced. The development and growth of muscles is a complex process regulated by endocrine hormones [5,6], growth factors, and signaling pathways [7,8]. The AMPK signaling pathway plays a crucial role in maintaining energy and lipid metabolism balance and regulates muscle growth, size, and hypertrophic capacity [9]. Phosphorylated AMPK modulates the lipogenic transcriptional factor SREBP1 and acetyl-CoA carboxylase (ACC1 and ACC2), stimulating fatty acid biosynthesis and signaling to LPL to release fatty acids [10]. Concurrently, phosphorylated AMPK (P-AMPK) activates CPT1A, promoting fatty acid oxidation and energy production [11]. These signals are transmitted to CD36, which transports fatty acids into cells for uptake and utilization, supporting muscle growth and regulating muscle size and hypertrophic capacity [12]. However, the specific regulatory mechanisms of muscle growth and differentiation between different sexes are not yet clear.

Therefore, we hypothesize that muscle development in different sexes of Muscovy ducks is regulated by the hormone and AMPK signaling pathway. In this study, we measured and analyzed various growth traits and conducted muscle transcriptome sequencing on male and female ducks to identify the effects of these genes and pathways on muscle tissue development. By analyzing the mechanisms of breast muscle growth differences in different sexes of Muscovy ducks, we aim to improve the breast muscle development level of female ducks, narrow the gender gap, increase the market value of female ducks, and reveal the basic physiological and genetic mechanisms of avian muscle growth, providing important data and theoretical support for animal science and livestock research.

## 2. Results

### 2.1. Analysis of the Growth and Development Patterns of Muscovy Ducks

The average weekly weight (Figure 1 A) and the curve of weekly weight gain (Figure 1B) indicate that both male and female ducks exhibit a slower growth rate in the first 3 weeks of life, followed by accelerated growth from 3 to 10 weeks, reaching peak weight gain between 5 and 7 weeks. Weight gain gradually declines after reaching 7 weeks of age. Notably, starting from week 2 to 3, male ducks demonstrate higher weight gain compared to female counterparts.

### 2.2. Measured Growth Traits and Serum Hormone Levels of Male and Female Muscovy Ducks

The normality and homogeneity of variance in the growth traits of 10 weeks old male and female Muscovy ducks were assessed, as detailed in Appendix A. The findings indicated no differences among the datasets (*p* > 0.05), demonstrating high reliability. A difference analysis (Figure 2) was conducted for each growth trait, and it was found that the semi-immersion length, body slope length, keel length, hip bone width, and chest width of the male ducks were greater than female ducks (*p* < 0.01). Moreover, these measurements were all positively correlated with body weight (*p* < 0.01).

Compared to the male group, the serum testosterone level of the 10-week-old female group is decreased (*p* < 0.01), while the serum estradiol levels are increased (*p* < 0.01, Figure 3).

### 2.3. RNA-seq of Muscle Tissues from Male and Female Domestic Muscovy Ducks

RNA-seq was conducted to investigate influence of genes and pathways on Muscovy muscle tissue development. Five Muscovy ducks were used to create a biological sample bank with three male and three female replicates. The raw reads exhibit high read quality (Appendix A). Each sample demonstrates approximately 60% alignment to the mallard duck reference genome, yielding a minimum total of 38,623,324 filtered reads (Appendix A). A total of 17,912 genes were identified, comprising 15,748 known genes and 2164 novel genes. Among these, 151 genes were differentially expressed, with 102 genes upregulated and 49 genes downregulated (Figure 4A, Appendix A). The heatmap clustering of genes with similar expression patterns effectively displays the differences in gene expression between male and female ducks (Figure 4B). To validate the RNA-seq results, we randomly selected three upregulated genes (*IGF1*, *GRIN3A*, *VIPR2*) and three downregulated genes *AMH*, *PAK1*, *SSTR2*) from breast muscle tissue (Figure 4C). This validation underscores the high reliability of the transcriptome findings presented in this study.

#### GO Function Annotation and the KEGG Pathway Analysis

In the GO analysis, a total of 451 clustered biological processes were enriched (Appendix A, *p* value < 0.05). Among them, enriched terms closely associated with muscle growth and development can be divided into five categories (*p* adjust < 0.05): muscle myosin complex, positive regulation of skeletal muscle acetylcholine-gated channel clustering, physiological muscle hypertrophy, skeletal muscle satellite cell maintenance involved in skeletal muscle regeneration, and regulation of skeletal muscle acetylcholine-gated channel clustering. (BP) A total of 332 entries were enriched in the biological process category, (CC) 39 entries were enriched in the cellular component category, and (MF) 80 entries were enriched in the molecular function category; the top 10 entries can be found in Figure 5A.

In the KEGG pathway analysis (Figure 5B), a total of 52 signaling pathways were enriched, and 21 signaling pathways (*p* adjust < 0.05) were enriched with multiple differentially expressed genes. In the top ten differentially enriched pathways, the AMPK signaling pathway, cAMP signaling pathway, and MAPK signaling pathway all related to growth metabolism and showed highly significant differences and gene enrichment between the male and female Muscovy duck. Notably, the AMPK signaling pathway is among the top 10 differential entries in the KEGG database (*p* adjust < 0.05).

### 2.4. The Expression Level of the AMPK Pathway in Male and Female Muscovy Ducks

The mRNA expression levels of *CD36*, *CPT1A*, and *LPL* in the mother duck group were higher than those in the male duck group (*p* < 0.05). Conversely, the expression levels of *ACC1* and *AMPK* mRNA in the female group were comparatively elevated compared to those in the male group; however, no statistically significant difference was observed between the two groups (*p* > 0.05, Figure 6).

Additionally, the P-AMPK protein expression level in female ducks was lower than male ducks (*p* < 0.05), while the CD36 and SREBP1protein expression level in female ducks were higher (*p* < 0.05) than male ducks; there were no differences in the expression levels of AMPK proteins (Figure 7).

## 3. Discussion

The Muscovy duck’s substantial meat production potential and quality advantages make it an important component of meat consumption in China. Notably, compared to domestic ducks, Muscovy ducks have higher breast and leg muscle meat yields and lower fat content [13]. However, the growth rate and feed conversion rate of male Muscovy ducks are much higher than those of female ducks, and at 60 days of age, male Muscovy ducks weigh 1.45-times more than female ducks [4]. This gender disparity results in production inefficiencies, as male and female ducks are unable to develop evenly. Consequently, this imbalance adversely affects the scalability and profitability of the meat duck industry. Therefore, in this study, exploring the muscle development patterns in male and female ducks can help narrow the gender gap, improve the growth performance of female ducks, and promote the sustainable development of the Muscovy duck meat industry.

Analyzing growth development curves enables the direct identification of growth trends [14]. This approach allows for the direct observation of critical growth milestones in animals, helping to determine when body weight typically stops increasing significantly [15]. This indicates that muscle growth has already approached its maximum potential [7]. Additionally, growth traits, such as body weight, body depth, half-immersion length, keel length, chest width, chest depth, and hip bone width, are essential indicators that comprehensively reflect an animal’s growth and development [16,17]. In this study, we fitted the weight changes in male and female Muscovy ducks from 0 to 10 weeks. At 10 weeks, the weight difference between males and females was found to be the greatest, and the weight growth rate tended to level off. Furthermore, we observed that the growth traits of male Muscovy ducks were significantly greater than those of female Muscovy ducks. These measurements were all positively and significantly correlated with body weight, which is consistent with findings from our previous studies [18]. Previous studies have found that at 10 weeks of age, the abdominal fat of Black Muscovy ducks accounts for only 1% of the total eviscerated weight, while the muscle accounts for 30% [19]. Furthermore, in current Muscovy duck production practices, 10 weeks is typically considered the market age for meat ducks. This indicates that the primary reason for the weight difference in 10-week-old ducks is the muscle weight. Therefore, it was determined that at 10 weeks, the body development of male and female ducks tends to be complete, and muscle development tends to be mature. This makes the 10-week mark an ideal time to study the weight differences between male and female ducks, with a particular focus on muscle development. By establishing that body and muscle development are nearly complete at this age, we can better understand the growth dynamics and optimize production practices based on these insights.

Sex hormones, such as testosterone, play a pivotal role in regulating muscle development. These hormones influence various physiological processes that contribute to muscle growth, repair, and maintenance [20]. Supplementation of exogenous testosterone in males results in a substantial enhancement of muscle protein synthesis, with evidence indicating that testosterone can promote muscle hypertrophy through upregulation of muscle protein synthesis [21]. Older men who take testosterone for three years experience significant improvements in muscle mass and quality [22]. These findings suggest that these hormones play a significant role in muscle growth, repair, and maintenance. However, the precise regulatory mechanisms by which they exert these effects remain to be fully elucidated and warrant further research.

Testosterone can indeed regulate the AMP-activated protein kinase (AMPK) pathway, thereby influencing muscle growth and development [23]. AMPK is a crucial energy-sensing enzyme involved in regulating cellular energy balance and metabolic processes. Cellular levels of AMP and ADP rise are regulated by hormonal levels. Upstream kinases such as serine/threonine kinase (LKB1) and calcium/calmodulin-dependent protein kinase kinase 2 (CAMKK2) activate AMPK in response to these increased levels of AMP and ADP. Once activated, phosphorylated AMPK leads to the negative feedback regulation of key lipogenic genes such as *SREBP1* [10,24], *CD36* [25], and *CPT1A* [24], thereby promoting muscle cell growth, development, and regeneration [26,27]. In addition, increased LPL (lipoprotein lipase) activity in skeletal muscle aids in lowering plasma triglyceride levels and enhances the uptake of free fatty acids into muscle cells [28]. These metabolic adjustments not only optimize energy use in muscle tissue but also contribute to overall body weight gain.

Determining which processes support muscle growth and regulate muscle size and the capacity for hypertrophy maintenance in muscle tissue is important [9]. In this study, the AMPK signaling pathway was identified as one of the top ten differentially enriched pathways in the KEGG analysis of male and female duck pectoral muscle tissues. Subsequent validation using qPCR and Western blotting revealed that the activation level of the AMPK pathway was higher in the pectoral muscle of male ducks. Therefore, we infer that the differences in muscle development between male and female ducks are primarily regulated by the activation of the AMPK pathway under the influence of sex hormones.

## 4. Materials and Methods

### 4.1. Experimental Animal Selection

A total of 200 healthy, white-feathered Muscovy ducks, comprising 100 males and 100 females, were selected at 0 days old and housed in a commercial, confined duck facility at Wen’s Muscovy Duck Incubation Base (Wen’s Food Group Co., Ltd., Jiangle County, Sanming City, Fujian Province, China). The ducklings were reared in a barn with consistent temperature, humidity, and lighting settings. The ducks had ad libitum access to feed and water and were provided with a commercial pellet diet without antibiotics twice daily. All experimental ducks underwent routine immunization, and the barn was disinfected weekly.

### 4.2. Growth Traits and Tissue Sample Collection

All of the 100 male and 100 female Muscovy ducks were reared and weighed weekly. At 10 weeks of age, 15 male and 15 female ducks were selected randomly to two experimental groups, namely the male group and the female group, to measure seven growth traits: body weight, body depth, half-immersion length, keel length, chest width, chest depth, and hip bone width. Then, 15 male ducks and 15 female ducks were selected for humane euthanasia via cervical dislocation, and muscle samples were collected. To reduce individual variations, five Muscovy ducks were combined in equal numbers to form a biological sample bank, resulting in three biological replicates for the male (Male1, Male2, and Male3) and female groups (Female1, Female2, and Female3).

### 4.3. Serum Hormone Assay

At 10 weeks of age, blood samples were collected from 15 male and female ducks. Approximately 500 μL of blood was drawn into an anticoagulant tube and stored at −20 °C for subsequent analysis. Serum levels of testosterone and estrogen were quantified using an ELISA kit (ZCIBIOTechnology, Shanghai, China).

### 4.4. Library Construction and Sequencing

Total RNA was extracted from ovarian tissue using TRIzol^®^. RNA concentration and purity were measured with a NanoDrop 2000 Spectrophotometer (Thermo Fisher Scientific, Waltham, MA, USA), and integrity was assessed with a 2100 Bioanalyzer and RNA Nano 6000 Assay Kit (Agilent Technologies, Santa Clara, CA, USA). mRNA libraries were created using the Illumina TIANGEN Biotech platform (TIANGEN Biotch Co., Ltd., Beijing, China) and sequenced by the Illumina Novaseq/Xplus. Reads were aligned to the reference genome with HISAT2 software (version 2.1.0). The software DESeq2 (version 0.6.0) [29] and edgeR (version 3.32.1) [30] was used to identify differentially expressed transcripts. GO analysis was performed using topGO (version 2.36.0) and clusterProfiler (version 3.12.0) for KEGG functional enrichment analysis. The raw sequencing data have been placed in the NCBI Sequence Read Archive within the scope of the BioProject (PRJNA1140738).

### 4.5. RT-qPCR Validation of Differentially Expressed Genes

Total RNA was extracted from tissues using the Trizol reagent (Invitrogen, Carlsbad, CA, USA) following the manufacturer’s instructions. The RT-qPCR analysis was conducted according to our previous study [31]. The quality of the RNA sample was assessed using 1% agarose gel electrophoresis, and the purity of PCR products was determined using a nucleic acid analyzer (NanoDrop 1000; Thermo Fisher Scientific, Waltham, MA, USA). The first-strand cDNA was synthesized with the Evo M-ML V kit from ACCURATE BIOLOGY (Hangzhou, Zhejiang, China). Quantitative real-time PCR analysis for each gene and sample was conducted using the HEAL FORCE real-time PCR system (CG-02/05, Shanghai, China), with three measurements performed for each. All genes were normalized to GAPDH and calculated using the 2^−ΔΔCT^ method. The primer sequences for this experiment can be found in Appendix A.

### 4.6. Western Blot Analysis for Muscovy Ducks’ Muscle Tissue

The protein was isolated from muscle tissue using a total protein extraction kit (Epizyme Biomedical, Shanghai, China), and its concentration was assessed with a BCA quantitation kit (Epizyme Biomedical, Shanghai, China). The separation gel was prepared utilizing an SDS-PAGE kit (Epizyme Biomedical, Shanghai, China). The proteins were subsequently eluted from the SDS-PAGE gel and transferred onto a PVDE membrane. Subsequently, the membrane was incubated at room temperature for 2 h in 3% BSA, followed by overnight exposure to primary antibodies at 4 °C and subsequent incubation with secondary antibodies at room temperature in the dark. The primary antibodies used included AMPK (HUAAN, Hangzhou, Zhejiang, China; dilution: 1:1000), phosphorylation of AMPK/P-AMPK (Zenbio, Chengdu, Sichuan, China; dilution: 1:1000), CD36 (Zenbio, Chengdu, Sichuan, China; dilution: 1:1000), SREBP1 (Zenbio, Chengdu, Sichuan, China; dilution: 1:1000), and GADPH (Proteintech, Wuhan, Hubei, China; dilution: 1:2000). After being washed three times with 1 × TBST, the membranes were incubated with the secondary antibody (goat anti-rabbit IgG (H + L) HRP, Abcam, UK, 1:10,000) for 1 h. The membranes were subsequently washed twice with 1 × TBST and once with 1 × PBS for 5 min each. Subsequently, the membranes were subjected to incubation with an ECL kit (Advansta Inc., San Jose, CA, USA) following the manufacturer’s protocol. Chemiluminescence images of the membranes were captured using an analyzer (Baygene Biotechnology, Beijing, China) and analyzed in grayscale with Image J software (version 1.52).

### 4.7. Statistical Analysis

Growth traits, serum hormone levels, and mRNA and protein expression data were analyzed using the Student’s *t*-test, with results presented as mean ± standard error of the mean (SEM) using SPSS software (version 29.0). Weight measurements for ducks aged 0–10 weeks were plotted using Excel 365 (version 4.3.4.28). RNA-seq gene expression levels were evaluated using HT-Seq software (version 0.6.0). Adjusted *p*-values (*p*. adj) were employed to control the false discovery rate in hypothesis testing. Criteria for selecting differential genes included an absolute log2 fold change greater than 1 and a *p*-value less than 0.05. Bar charts were utilized for both Gene Ontology (GO) and Kyoto Encyclopedia of Genes and Genomes (KEGG) analyses. Statistical significance was established at *p* < 0.05, denoted with *, *p* < 0.01, denoted with **, and *p* < 0.001, denoted with ***. Graphpad prism (version 10.1) was used to create visualizations.

## 5. Conclusions

In conclusion, by combining the analysis of weight, growth traits, and transcriptome sequencing of muscle tissues in male and female ducks, alongside the validation of key signaling pathways, we inferred that the growth and development of pectoral muscles in male and female ducks are regulated by sex hormones. This regulation is likely mediated through the modulation of the AMPK signaling pathway.

## Figures and Tables

**Figure 1 ijms-25-10132-f001:**
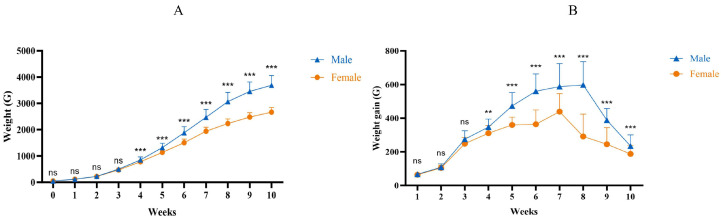
Weight changes in male and female Muscovy ducks at different ages. (**A**): Growth curves. (**B**): Weekly weight gain curve. Female: the female Muscoy duck group. Male: the male Muscoy duck group. “**” indicates *p* < 0.01; “***” indicates *p* < 0.001; “ns” indicates *p* > 0.05.

**Figure 2 ijms-25-10132-f002:**
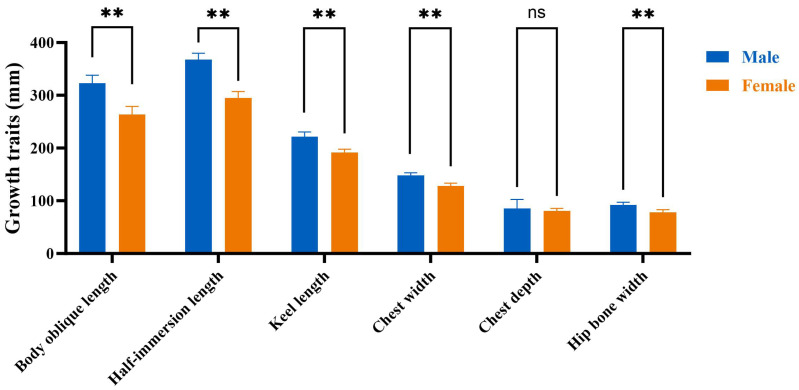
Comparison of growth traits between male and female Muscovy ducks. Male represents male Muscovy ducks. Female represents female Muscovy ducks; “**” represents *p* < 0.01; “ns” represents *p* > 0.05.

**Figure 3 ijms-25-10132-f003:**
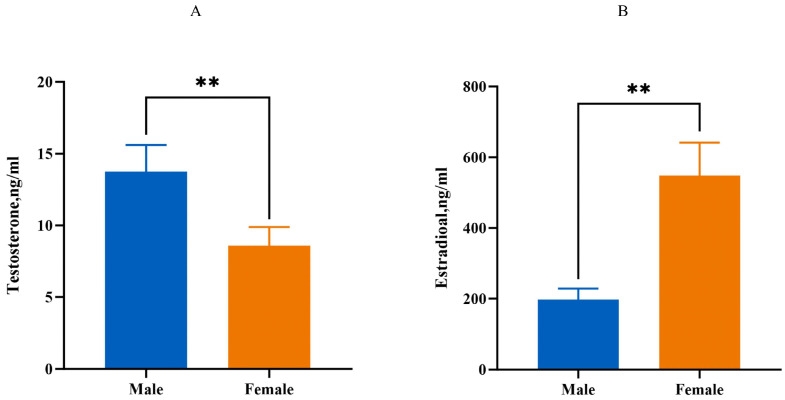
Estimation of serum estrogen and testosterone levels. (**A**): The level of testosterone. (**B**): The level of estrogen. Male: male Muscovy duck group. Female: female Muscovy duck group. “**” represents *p* < 0.01.

**Figure 4 ijms-25-10132-f004:**
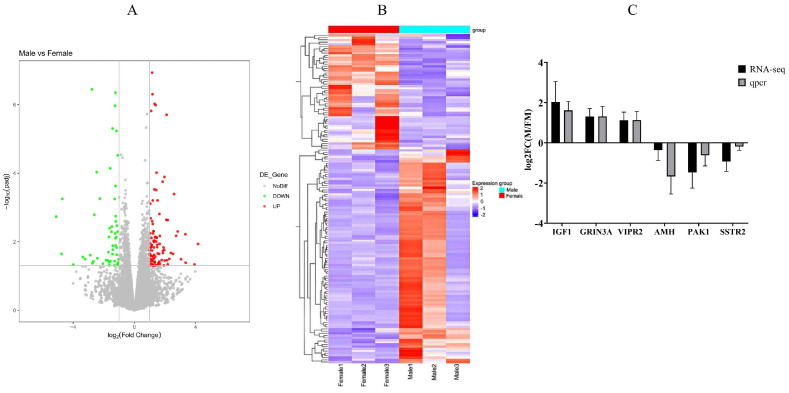
Genetic variation prediction and expression analysis. (**A**): A volcano plot of gene differential expression. (**B**): Cluster analysis of differential genes. (**C**): qRT-PCR verification of differentially expressed genes. RNA-seq: gene expression trends of RNA-seq. qPCR: qPCR gene expression trends. Male: male Muscovy duck group. Female: female Muscovy duck group.

**Figure 5 ijms-25-10132-f005:**
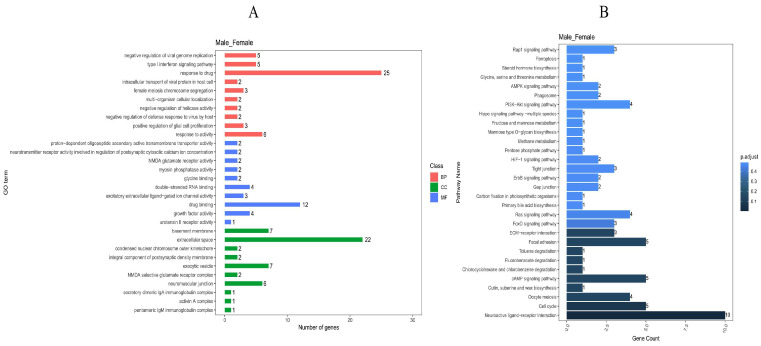
GO and KEGG enrichment analysis of differential genes. (**A**): GO enrichment entries of male and female group. (**B**): KEGG enrichment entries of male and female group.

**Figure 6 ijms-25-10132-f006:**
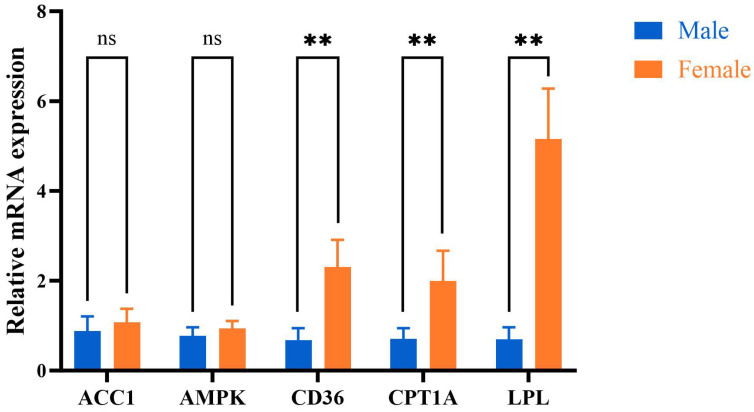
The mRNA expression levels of AMPK signaling pathway. Male: male Muscovy duck group. Female: female Muscovy duck group. “** “represents *p* < 0.01; “ns “represents *p* > 0.05.

**Figure 7 ijms-25-10132-f007:**
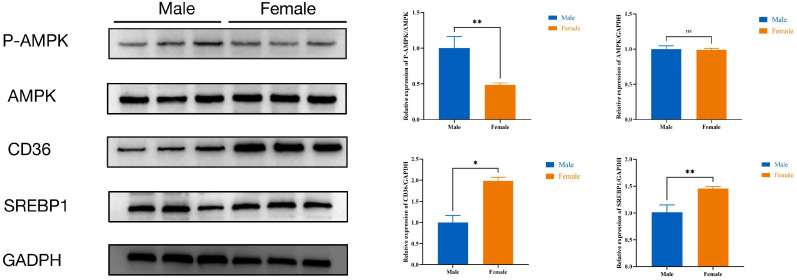
The protein expression levels of AMPK signaling pathway. Male: male Muscovy duck group. Female: female Muscovy duck group. “*” represents *p* < 0.05, “**” represents *p* < 0.01, and “ns” represents *p* > 0.05.

## Data Availability

Data is contained within the article and Appendix A.

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
