# Peer review of "Involvement of the AMPK Pathways in Muscle Development Disparities across Genders in Muscovy Ducks"

_ijms, 2024, doi:10.3390/ijms251810132_

Round 1
Reviewer 1 Report
Comments and Suggestions for Authors
In this study, the authors measured the growth traits, serum hormone levels and analyzed muscle RNA-seq on male and female ducks to identify the genes and pathways involved in muscle tissue development. They found that the weight difference between male and female ducks is likely mediated through the activation of AMPK signaling pathway. However, they did not conduce any function study to confirm this hypothesis, and that is why the authors used the word of “likely”, which makes the conclusion of this manuscript is ambiguous.
Other comments:
Figure 1A and 1B, add standard deviation and P value of statistic analysis using asterisk.
Section 2.2, add age information for growth trait measurement in the main text.
Section 2.3, the authors should add the purpose of doing RNA-seq in the first place and add the animal numbers used for RNA-seq.
Figure 4C, add standard deviation.
Section 2.4, explain why checked CD36, CPT1A, and LPL 3 genes?
In addition, there are a few writing format issues are needed to modify.
Reviewer 2 Report
Comments and Suggestions for Authors
The topic is interesting. The experimental design is correctly described and could be repeated. Methods are relevant.
Specific comments:
- L279: Statistical cinstead of Stiatistical
- There is the phrase: 'the same below' under figures. What does it mean?
- There is n=6 below figure 4-7. What does this number mean? In Abstract and M&M section, sample size was at the beginning 100-100 and 15-15 in 4.2. Sampling and sample size should be clarified.
- I suggest Abstract in a single paragraph, the present broken version is a little bit hard to understand.
- The title and keywords are almost the same. Keywords should be changed to give more information.
Round 2
Reviewer 1 Report
Comments and Suggestions for Authors
The authors have addressed all my comments, I do not have additional questions.
Reviewer 2 Report
Comments and Suggestions for Authors
The paper is significantly improved.
I only have problem still with the chosen keywords. The title of the manuscript is: Involvement of the AMPK Pathways in Muscle Development Disparities Across Genders in Muscovy Ducks, while keywords are: Muscovy ducks; growth traits; hormone; AMPK. It would be better to use keywords not parts of the title. I ask the authors thinking on it for a while.